# Model Tells Itself Where to Attend: Steerable Prompting for Reliable Reading Comprehension of LLMs

## Abstract

Large language models (LLMs) have demonstrated remarkable performance across various real-world tasks. However, they often struggle to fully comprehend and effectively utilize their input contexts, resulting in responses that are hallucinated. This difficulty increases for contexts that are long or contain distracting information, which can divert LLMs from fully capturing essential evidence. To address this issue, many works use prompting to help LLMs comprehend contextual information more reliably. For instance, iterative prompting highlights key information in two steps that first ask the LLM to identify important pieces of context and then derive answers accordingly. However, textual prompting methods are constrained to highlighting key information implicitly in token space, which is often insufficient to fully steer the model's attention. To improve model reading comprehension, we propose SteerPrompt, a method that automatically identifies key contextual information and explicitly highlights it by steering an LLM's attention scores. Like prompting, SteerPrompt is applied at inference time and does not require changing any model parameters. Our experiments on open-book QA demonstrate that SteerPrompt effectively enables models to grasp essential contextual information, leading to substantially improved problem-solving performance, e.g., an average improvement of 7.95% for LLAMA3-70B-Instruct. Code will be publicly available.

## 1 Introduction

Large language models (LLMs) exhibit remarkable performance across various natural language processing (NLP) tasks and artificial intelligence (AI) applications (Brown et al., 2020; Touvron et al., 2023; OpenAI, 2023). Despite their remarkable capabilities, recent studies reveal that LLMs often encounter challenges in fully understanding their input contexts, overlooking or showing insensitivity to crucial contextual information (Kasai et al., 2023; Li et al., 2023; Si et al., 2023; Zhou et al., 2023; Yu et al., 2024; Zhang et al., 2024). Consequently, the models tend to fabricate answers (also known as hallucination), resulting in responses that are inconsistent with the presented contexts (Zhou et al., 2023; Yu et al., 2024). This becomes particularly problematic when models are presented prompts containing lengthy background contexts (Liu et al., 2023) or complex questions, such as in open-book question answering (QA) (Kwiatkowski et al., 2019; Shi et al., 2023b; Peng et al., 2023). In these information-dense scenarios, lengthy contexts can overwhelm LLMs, which contain many details with varying degree of relevance (Wan et al., 2024; Zhang et al., 2024). Some sentences are crucial for providing the correct answer, while others, though irrelevant, can distract models from fully understanding the essential information.

To improve reading comprehension of models, most prior work explores well-designed prompts to guide the LLM to use contextual knowledge more reliably (Zhou et al., 2023; Wan et al., 2024; Radhakrishnan et al., 2023). In particular, *iterative prompting* in chain-of-thought (COT; Wei et al., 2022) fashion can help LLMs decompose complex task-solving into more interpretable and manageable intermediate steps, thus yielding better performance (Radhakrishnan et al., 2023). Motivated by this, it is natural to design multi-step iterative prompting to guide LLMs to pay more attention to relevant contextual parts and derive answers accordingly. Specifically, for open-book QA tasks, iterative prompting can be decomposed into two steps: (i) *identifying key information* and (ii) *deriving answers using the key information*.

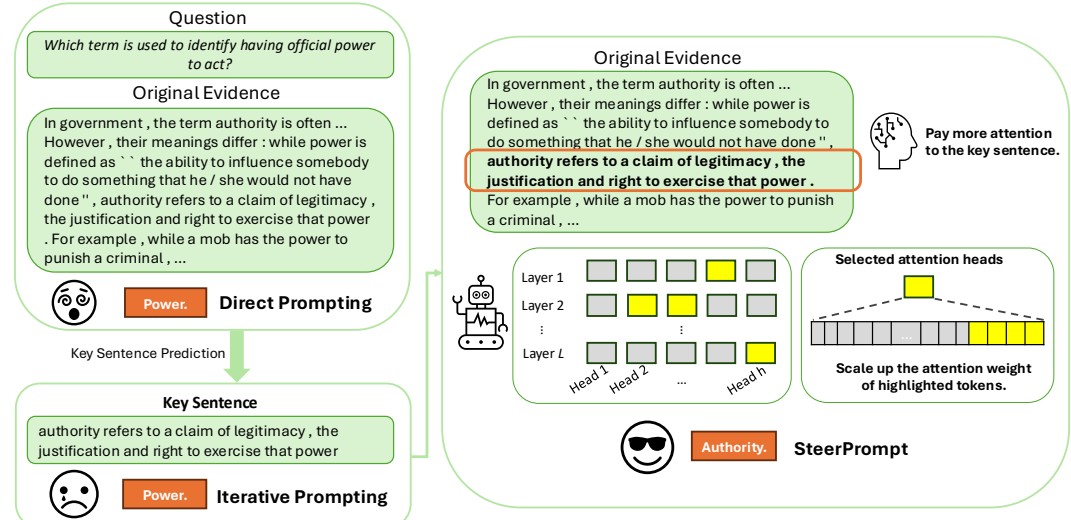

Figure 1: The illustration of SteerPrompt and alternative methods given a running example. Responses by Vicuna-7B are shown in red square where *Authority* is the label. Prompting methods (direct and iterative prompting) fail to guide a model to derive correct answers while SteerPrompt successfully steers it to answer correctly by explicitly highlighting identified key parts.

Iterative prompting can work effectively for black-box LLMs of significantly large sizes (e.g., >100B) (Radhakrishnan et al., 2023). However, for LLMs of smaller sizes (e.g., LLAMA3-70B, Meta, 2024), it remains unclear if this strategy can guide models to fully attend to the extracted key information and subsequently improve performance. First, step-by-step generations typically result in longer contexts. However, key information is only highlighted in token space by appending the short predicted key sentences, which are often not strong enough to fully steer the model's attention. As illustrated in the left part of Figrue 1, even though the model correctly predicts the key sentence which is appened in the subsequent prompt, it still fails to provide the correct answer. Moreover, errors can propagate across steps, further compromising performance. Therefore, we aim to develop an alternative inference method that emulates iterative prompting while addressing these limitations.

Motivated by this, we propose SteerPrompt, an inference-only method that (i) *automatically* identifies key contextual parts, and (ii) *explicitly* highlights them through attention score manipulation for improving models' reading comprehension and performance on open-book QA tasks. Specifically, SteerPrompt integrates iterative question-decomposition prompting and attention steering approaches (Zhang et al., 2024). Given the original context and question, SteerPrompt first prompts an LLM to identify the key information (sentences) through free-text generation. Then, instead of appending those key sentences to the initial prompt, SteerPrompt maps those sentences back to the original context using semantic embeddings (Figure 1 right). By using the original sentences for highlighting, it avoids more lengthy input for the next step, and potentially reduces the unreliable key sentences generations, mitigating the error propagation. Finally, to guide the model to attend to the selected key sentences, SteerPrompt highlights them through attention steering that upweights their corresponding attention scores at the selected attention heads as done by Zhang et al. (2024). Unlike existing attention steering work, our method does not necessitate human annotation on the highlighting part, rectifying its critical limitation. Additionally, we also design an efficient coarse-to-fine search scheme for identifying effective attention heads for steering, which reduces the searching overhead by 4.5× compared to the greedy method of previous work (Zhang et al., 2024).

We conduct experiments to evaluate the effectiveness of SteerPrompt using Vicuna-7B (Chiang et al., 2023), LLAMA3-8B-Instruct, and LLAMA3-70B-Instruct (Meta, 2024) on both single- and multi-hop open-book QA tasks from Natural Questions (NQ; Kwiatkowski et al., 2019) and Hot-potQA (Yang et al., 2018b). Empirical evidence shows that it can be a much easier task for the LLMs to select the contextual key sentences than comprehending the full context (see analysis in Section 5.1). SteerPrompt converts the challenging bottleneck of contextual comprehension into an easier problem of key-sentence selection in a steerable way, consistently providing significant performance improvements over baseline prompting strategies. For example, SteerPrompt achieves an average improvement of 8.99% on exact-match (EM) score over iterative prompting for LLAMA3-

70B-Instruct across both tasks. Remarkably, the attention head sets obtained by SteerPrompt exhibit outstanding generalization ability, allowing them to be effectively used across different tasks.

## 2 BACKGROUND

**Problem description.** In standard LLM prompting, we are given a pre-trained LLM and a text prompt $x$ consisting of $n$ tokens. In the closed-book setting, the prompt $x$ can only be a question or instruction to be completed by models. Relying solely on model parametric knowledge poses challenges in scenarios involving complex questions that entail new knowledge or private information (Zhou et al., 2023; Yu et al., 2024). Existing methods (Shi et al., 2023b; Peng et al., 2023) resort to augmenting the prompt with additional background contexts to facilitate question answering, i.e., open-book question answering. The following box presents a prompt template that we use for open-book QA:

> **A direct prompt template for open-book QA**
>
> Answer the question below, paired with a context that provides background knowledge. Only output the answer without other context words.
>
> Context: {context}
>
> Question: {question}
>
> Answer:

**Multi-head attention.** A typical transformer model consists of $L$ stacked layers, where each layer contains two submodules: a multi-head attention (MHA) and a fully connected feed-forward network (FFN). Given the input $\boldsymbol{X} \in \mathbb{R}^{n \times d}$, MHA of the layer $l$ performs the attention function in parallel $H$ heads: $\mathrm{MHA}^{(l)}(\boldsymbol{X}) = \mathrm{Concat}(\boldsymbol{H}^{(l,1)}, ..., \boldsymbol{H}^{(l,H)})\boldsymbol{W}_o$ with

$$\boldsymbol{H}^{(l,h)} = \mathrm{Softmax}(\boldsymbol{A}^{(l,h)})\boldsymbol{V}^{(l,h)}$$

where $\boldsymbol{A} = \frac{1}{\sqrt{d_h}}\boldsymbol{Q}\boldsymbol{K}^\top \in \mathbb{R}^{n \times n}$ is the scaled inner product between query $\boldsymbol{Q}$ and key $\boldsymbol{K}$. $\boldsymbol{Q} = \boldsymbol{X}\boldsymbol{W}_{q_h}, \boldsymbol{K} = \boldsymbol{X}\boldsymbol{W}_{k_h}, \boldsymbol{V} = \boldsymbol{X}\boldsymbol{W}_{v_h}$ and $\boldsymbol{W}_{q_h}, \boldsymbol{W}_{k_h}, \boldsymbol{W}_{v_h} \in \mathbb{R}^{d \times d_h}$ are learnable projection matrices of head $h$. $d_h$ is typically set to $d/H$.

**Post-hoc attention steering.** Zhang et al. (2024) propose PASTA, an inference-only method that applies attention reweighting to steer model attention towards user-highlighted input sets, thereby improving instruction following and contextual comprehension. Specifically, given the index set of user-specified tokens as $\mathcal{G}$ ($\mathcal{G} \subset [n]$), PASTA highlight these tokens by upweighting their attention scores with a constant attention bias $\boldsymbol{B}^{(l,h)}$:

$$\boldsymbol{H}^{(l,h)} = \mathrm{Softmax}(\boldsymbol{A}^{(l,h)} + \boldsymbol{B}^{(l,h)})\boldsymbol{V}^{(l,h)}, \quad \boldsymbol{B}^{(l,h)}_{ij} = \left\{ \begin{array}{ll} -\delta & \text{if } (l,h) \in \mathcal{H} \text{ and } j \notin \mathcal{G} \\ 0 & \text{otherwise.} \end{array} \right. \quad (1)$$

where $\delta$ is a positive constant. After $\mathrm{Softmax}(\cdot)$, the attention scores of tokens not in $\mathcal{G}$ is scaled down by $\exp(\delta)$. Correspondingly, the others in $\mathcal{G}$ are upweighted due to the renormalization of Softmax[1], steering the model to pay more attention to the input spans of $\mathcal{G}$. Following Zhang et al. (2024), we set $\delta = \log 100$ in all of our experiments. $\mathcal{H}$ is an index set of attention heads selected for steering. Since various heads function diversely, steering different heads yields dramatically different performance. To identify the effective heads, Zhang et al. (2024) employ a greedy search approach that evaluates the steering performance of each head on multiple tasks and selects those with best accuracy. The resulting head set $\mathcal{H}$ can be generalized for steering across different tasks.

PASTA requires the access to user-annotated input spans for highlighting. In the case of context-specific tasks, it is generally prohibitively expensive to extract and annotate relevant sentences from lengthy contexts through humans. To address this critical limitation and improve model reading comprehension by automatic explicit highlighting, we introduce our method – SteerPrompt.

---

[1](1) is a simplified formula from Equation (2) in Zhang et al. (2024), which we elaborate in Appendix A.

## 3 METHOD

Our proposed method – Steerable Prompting (SteerPrompt), integrates iterative prompting and attention steering. This integration synergistically combines the advantages of both techniques while mitigating their respective limitations. For multi-step iterative prompting, incorporating attention steering externalizes the highlighting of key information through an inference-only operation, efficiently enhancing model reading comprehension with improved reliability and controllability. For post-hoc attention steering, equipping it with iterative prompting enables the automatic identification of contextually relevant key information, thereby addressing its significant reliance on human annotations.

---

**Algorithm 1** SteerPrompt

---

**Input** a question $q$, a context $c$, the head set $\mathcal{H}$ of an LLM $\mathcal{M}$, prompt templates $\mathcal{P}_i, \mathcal{P}_d$ and $\delta$.
  1: Generate $g_1 = \text{Generate}_{\mathcal{M}}(\mathcal{P}_i(q, c))$ as in (2);
  2: Calculate $s_k = \text{Match}_e(g_1, \{s_1, \ldots, s_m\})$ as in (3);
  3: Steer $g_2 = \text{Steer}_{\mathcal{H}, s_k}(\text{Generate}_{\mathcal{M}}(\mathcal{P}_d(q, c)))$ as in (4);
**Output:** The final answer $g_2$

---

### 3.1 AUTOMATIC CONTEXTUAL HIGHLIGHTING

In the open-book QA task, an LLM $\mathcal{M}$ is prompted to answer a question $q$ paired with a background context $c$ that consists of $m$ sentences $c = s_1 || \ldots || s_m$. Instead of directly prompting an LLM with $(q, c)$, SteerPrompt first prompts the LLM to generate a key sentence from the context $c$ that supports answering the question:

$$g_1 = \text{Generate}_{\mathcal{M}}(\mathcal{P}_i(q, c)), \tag{2}$$

where $\mathcal{P}_i$ is the prompt template of key sentence identification that we show in Section 4.1. Then, SteerPrompt maps $g_1$ back to a sentence from the original context $c$ to avoid potential token-level generation errors in $g_1$ and mitigate error propagation. Specifically, it employs a small encoder $e$ to calculates the semantic embeddings of $g_1$ and every $s_i(1 \leq i \leq m)$, and pick the best-matching sentence $s_k$ with the highest similarity to $g_1$:

$$s_k = \text{Match}_e\Big(g_1, \{s_1, \ldots, s_m\}\Big) \subset c. \tag{3}$$

In the final step, SteerPrompt steers the attention scores of tokens in $s_k$ based on (1) at the specific attention heads $\mathcal{H}$, when directly prompting the LLM $\mathcal{M}$ to derive the answer for $(q, c)$:

$$g_2 = \text{Steer}_{\mathcal{H}, s_k}\Big(\text{Generate}_{\mathcal{M}}\left(\mathcal{P}_d(q, c)\right)\Big) \tag{4}$$

where $\mathcal{P}_d$ is the prompt template of direct answering as shown in Section 2, and $\text{Steer}_{\mathcal{H}, s_k}(\cdot)$ is detailed by (1) with $\mathcal{G}$ as the index set of $s_k$. As such, the identified key sentence $s_k$ is explicitly highlighted through attention score upweighting, directing the model to grasp the key information and solve the questions more reliably. Notably, SteerPrompt is applied at inference time and does not require changing any model parameters. More importantly, it does not involve human annotation on highlighted parts. The key information is automatically identified by iterative prompting the model $\mathcal{M}$, addressing the major limitation of existing attention steering approach.

### 3.2 COARSE-TO-FINE MODEL PROFILING

SteerPrompt requires carefully selecting $\mathcal{H}$, the set of attention heads to be steered in (1), but finding these heads can be computationally intensive. Zhang et al. (2024) propose a greedy search strategy that evaluates the steering performance of each head on small validation sets of multiple tasks and selects the heads that yield the best performance. This greedy strategy requires evaluating $L \times H$ times, resulting in non-trivial overheads especially for large models. To improve the efficiency of searching heads, we propose an alternative coarse-to-fine model profiling scheme that searches from the layer level to head level. Specifically, we first evaluate the performance of steering all attention heads of one single layer, then pick the top-$l$ layers, and further evaluate the steering performance of each head in these layers. The head set $\mathcal{H}$ is obtained by selecting the best-performing heads from top-$l$ layers. Empirically, we find that a small $l$ (e.g., $l = 6$ compared to $L = 32$) is sufficient for SteerPrompt to achieves superior performance and identify effective attention heads that can generalize across tasks, substantially reducing the searching overheads to $\frac{lH+L}{LH}$.

Table 1: Generation examples of a Vicuna-7B on NQ and HotpotQA. Texts in blue are predicted by the model for highlighting and texts in **bold** are highlighted by SteerPrompt.

| Task | Prompt | Baseline | SteerPrompt |
|------|--------|----------|-------------|
| NQ | Answer the question below, paired with a context that provides background knowledge. Only output the answer without other context words. Context: Although the delegates were divided early on as to whether to break from Crown rule, **the second Continental Congress on July 2, 1776, passed a resolution asserting independence, with no opposing vote recorded**. The Declaration of Independence was issued two days later declaring themselves a new nation: the United States of America. It established a Continental Army, giving command to one of its members, George Washington of Virginia. It waged war with Great Britain, made a militia treaty with France, and funded the war effort with loans and paper money. Question: when did the continental congress vote to adopt the declaration of independence? Answer: | ✗ The Continental Congress voted to adopt the Declaration of Independence on July 4, 1776. | ✓ July 2, 1776. Label: July 2, 1776. |
| HotpotQA | Answer the question below, paired with a context that provides background knowledge. Only output the answer without other context words. Context: [1]: Branford, Connecticut - **Branford is a shoreline town located on Long Island Sound in New Haven County, Connecticut, 8 mi east of New Haven**. The population was 28,026 at the 2010 census. [2]: Long Island Sound - **Long Island Sound is a tidal estuary of the Atlantic Ocean, lying between the eastern shores of Bronx County, New York City, southern Westchester County, and Connecticut to the north, and the North Shore of Long Island, to the south**. From east to west, the sound stretches 110 miles (177 km) from the East River in New York City, along the North Shore of Long Island, to Block Island Sound. A mix of freshwater from tributaries and saltwater from the ocean, Long Island Sound is 21 miles (34 km) at its widest point and varies in depth from 65 to. Question: How long is the tidal estuary in which Branford is a shoreline town? Answer: | ✗ Long Island Sound. | ✓ 110 miles. Label:110 miles. |

# 4 EXPERIMENTS

We conduct experiments to evaluate the effectiveness of SteerPrompt using Vicuna-7B (Chiang et al., 2023), LLAMA3-8B-Instruct, and LLAMA3-70B-Instruct (Meta, 2024) on both single- and multi-hop open-book QA tasks from Natural Questions (NA; Kwiatkowski et al., 2019) and HotpotQA (Yang et al., 2018b).

## 4.1 EXPERIMENTAL SETUP

**Datasets.** We study 2 datasets: HotpotQA (Yang et al., 2018a) and the MRQA version Fisch et al. (2019) of Natural Questions (NQ) (Kwiatkowski et al., 2019). Following the filtering procedures outlined by Yu et al. (2024), duplicated and low-quality questions are removed from the NQ dataset, resulting in 7,189 instances remaining in NQ, and 5,190 instances in HotpotQA. For each dataset, we randomly select 1,000 examples as the profiling set and keep the remaining examples as the test set (see breakdown in Table 7). For all the experiments, we present two evaluation metrics: Exact Match (EM), and Token-level F1 score. We apply greedy search decoding for all experiments.

**Implementation Details.** We implement our experiments using Huggingface Transformers (Wolf et al., 2019) and PyTorch (Paszke et al., 2019). All the experiments are conducted on NVIDIA A6000 and A100 GPUs.

**SteerPrompt Settings.** For SteerPrompt, we use the following prompt template $\mathcal{P}_i$ to prompt an LLM $\mathcal{M}$ to identify the key information from the context that support answering the question.

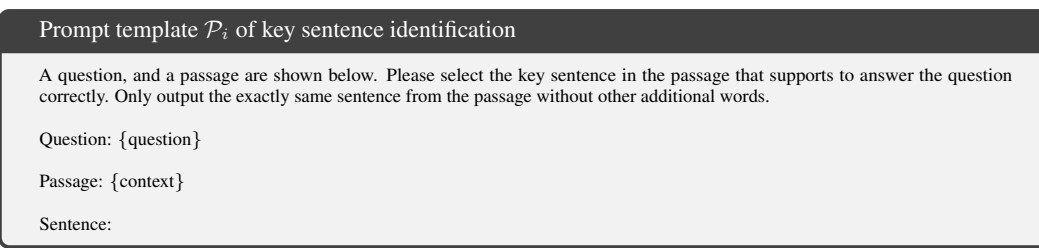

Prompt template $\mathcal{P}_i$ of key sentence identification

A question, and a passage are shown below. Please select the key sentence in the passage that supports to answer the question correctly. Only output the exactly same sentence from the passage without other additional words.

Question: {question}

Passage: {context}

Sentence:

Then, we map the predicted key sentence $\boldsymbol{g}_1$ back to the original context by (3), which uses a small encoder models to calculate the semantic embeddings of the predicted key sentence $\boldsymbol{g}_1$ and every

Table 2: Main evaluation results with Vicuna-7B, LLAMA-8B-Instruct, and LLAMA3-70B-Instruct on NQ and HotpotQA. "In-domain" means that the head set is selected based on the profiling set of the target task. "Out-of-domain" means that the head set is selected from the other dataset and the target task is unseen during the profiling.

| Model | Method | NQ | | HotpotQA | | All |
|---|---|---|---|---|---|---|
| | | EM | Token F1 | EM | Token F1 | Ave. |
| **Vicuna-7B** | Direct Prompting | 5.88 | 32.21 | 18.11 | 38.77 | 23.74 |
| | Iterative Prompting | 4.36 | 31.48 | 14.04 | 34.99 | 21.22 |
| | SteerPrompt$_{\text{out-of-domain generalize}}$ | 11.78 | 35.53 | 21.94 | 39.92 | 27.29 |
| | SteerPrompt$_{\text{in-domain profiling}}$ | 15.41 | 38.59 | 29.54 | 47.51 | 32.76 |
| **LLAMA3-8B** | Direct Prompting | 31.62 | 54.19 | 42.58 | 63.30 | 47.92 |
| | Iterative Prompting | 32.62 | 55.30 | 42.10 | 63.07 | 48.27 |
| | SteerPrompt$_{\text{out-of-domain generalize}}$ | 41.21 | 61.61 | 51.96 | 70.97 | 56.44 |
| | SteerPrompt$_{\text{in-domain profiling}}$ | 33.51 | 56.91 | 58.08 | 75.41 | 55.98 |
| **LLAMA3-70B** | Direct Prompting | 30.00 | 54.18 | 42.63 | 64.26 | 47.77 |
| | Iterative Prompting | 30.57 | 54.92 | 42.86 | 65.15 | 48.38 |
| | SteerPrompt$_{\text{out-of-domain generalize}}$ | 33.46 | 56.59 | 58.32 | 76.74 | 56.28 |
| | SteerPrompt$_{\text{in-domain profiling}}$ | 40.51 | 63.32 | 50.90 | 70.59 | 56.33 |

sentence $s_i$ in the context $c$. Specifically, we use a "all-MiniLM-L6-v2" model from Sentence-Transformer (Reimers & Gurevych, 2019) as the encoder to encode sentences. Then, we calculate the cosine similarity between semantic embeddings of $g_1$ and each sentence $s_i$ in the context, and select the contextual sentence $s_k$ with the highest similarity score as the final key sentence prediction. For multi-hop question answering, such as HotpotQA, the key sentences are identified for each individual hop separately. Finally, we highlight $s_k$ by (4) while directly prompting the model to answer the question paired by the context with the direct prompting template shown in Section 2.

**Coarse-to-fine Model Profiling.** For the coarse-to-fine search strategy outlined in Section 3.2, we consider all attention heads in the top-$l$ layers as potential candidates for selection, where $l$ is chosen from $\{3, 4, 5, 6\}$. Subsequently, we either select top-$i$ heads from each individual layer, or top-$j$ heads from the pool of head candidates. Top-$i$ is chosen from $\{4, 6, 8\}$, and top-$j$ is chosen from $\{16, 24, 32, 64\}$. The final head set utilized in the study is determined based on the highest token-F1 performance achieved on the profiling set.

**Baselines.** We evaluate three open-source LLMs: Vicuna-7B Chiang et al. (2023), Llama3-8B-Instruct, and Llama3-70B-Instruct under direct prompting, iterative prompting, and SteerPrompt.

- *Direct prompting:* Models are prompted to directly answer the question $q$ based on the provided context $c$. The prompt template $\mathcal{P}_d$ is displayed in Section 2.

- *Iterative Prompting*: Models are first prompted to generate the key sentence that supports answering the question, using the same prompt template $\mathcal{P}_i$. For multi-hop question answering, such as HotpotQA, the key sentences are identified for each individual hop separately (see Appendix C). The predicted key sentences are also mapped back to the original context, similar as that in SteerPrompt. Then, the model are prompted to answer the question with the key sentences appended to the context.

### 4.2 MAIN RESULT: STEERPROMPT IMPROVES OPEN-BOOK QA.

We evaluate the performance of SteerPrompt on NQ and HotpotQA in two settings: in-domain and out-of-domain evaluation. For the in-domain setting, we perform profiling (selecting the head set to steer) and evaluate performance on the same dataset. For the out-of-domain setting, we perform profiling and evaluate performance on different datasets, where the target task is unseen during the profiling to evaluate the generalization ability of SteerPrompt.

**In-domain Evaluation.** Table 2 suggests that, for all the models, SteerPrompt significantly improves the model performance compared with other baselines, regardless of model size and datasets. For

example, SteerPrompt achieves 40.51% EM for LLAMA3-70B-Instruct on NQ, yielding a significant 9.94% improvement compared to the best-performing baseline. We also observe that iterative prompting can mostly improve upon the direct prompting, showcasing the performance gains from identifying key sentences and appending them to contexts. However, in certain cases, such as Vicuna-7B, iterative prompting can actually underperform direct prompting. It suggests that highlighting in token space by appending key sentences is insufficient to fully steer a model's attention. In contrast, SteerPrompt shows a consistently substantial improvement over all baselines, demonstrating the effectiveness of automatic attention steering to improve model reading comprehension. Table 1 further illustrates this by comparing the generation examples of SteerPrompt and direct prompting.

**Out-of-domain Evaluation.** In this setting, given an evaluation task (e.g., NQ), we employ the head sets selected with profiling set of the other task (e.g., HotpotQA) for SteerPrompt to evaluate its generalization ability across different domains and tasks. The results in Table 2 indicate that SteerPrompt significantly outperforms all baseline methods for all models and all datasets, achieving better or comparable performance to that of in-domain profiling. Notably, for LLAMA3-8B-Instruct on NQ, the cross-domain performance surpasses the in-domain performance, compellingly demonstrating the robustness and generalization proficiency of our approach.

## 5 ANALYSIS

### 5.1 ISOLATING THE EFFECT OF STEERPROMPT'S TWO COMPONENTS

SteerPrompt consists of two primary components: automatic key sentence identification, and explicit highlighting key sentences. To underscore the necessity of both components, we conduct the comparison between following methods: (i) direct prompting with the original context; (ii) iterative prompting that appends the identified key sentences appended to the context; (iii) highlighting the entire context by attention steering approach but without key-sentence identification; (iv) SteerPrompt that highlights the identified key sentences. Moreover, we present additional results about iterative prompting to provide comprehensive evaluation for it. Specifically, within each context of the NQ dataset, there is one gold sentence that entails the answer. We also evaluate the performance of iterative prompting when appending the gold sentence.

The results in Table 3 indicate that SteerPrompt can benefit from using the identified key sentence, yielding significant performance gains. Specifically, highlighting the entire context via attention steering can improve upon direct prompting but underperforms SteerPrompt, suggesting the importance of key sentence identification. Meanwhile, the comparison between (ii) and (iv) illustrates the performance gains yielded by explicitly highlighting via attention steering. Therefore, these results suggest that both components are essential for SteerPrompt to achieve its best performance.

Table 3: Performance comparison among appending identified key sentences or gold sentence, and highlighting different parts of contexts. *N.A.* means that it cannot identify the gold sentence from HotpotQA that entails answers.

| Method | LLAMA3-8B on HotpotQA | | Vicuna-7B on NQ | |
|---|---|---|---|---|
| | EM | Token F1 | EM | Token F1 |
| Direct prompting | 42.58 | 63.30 | 5.88 | 32.21 |
| Iterative prompting w. identified key sentences | 42.10 | 63.07 | 4.36 | 31.48 |
| Iterative prompting w. gold sentences | *N.A.* | *N.A.* | 5.01 | 33.67 |
| Highlight the entire context | 54.88 | 73.23 | − | − |
| SteerPrompt highlights identified key sentences | 58.08 | 75.41 | 15.41 | 38.59 |

Moreover, we can see that, even appending the fully correct gold sentences to prompts, iterative prompting still faces challenges in effectively improving model's reading comprehension and QA performance. Therefore, the performance of prompting methods is bottlenecked by LLMs' capability of comprehending full contexts and grasping key information from them. By contrast, SteerPrompt addresses this bottleneck by explicitly highlighting key information through attention steering. As such, the QA performance will be upper bounded by the accuracy of key sentence identification. Notably, it is a much easier task for LLMs to select the contextual key sentences than comprehending

the full context. LLMs can achieve much higher accuracy of selecting key sentences than directly generating correct answers. Moreover, SteerPrompt proposes to map the key sentence generated by the model to a sentence from the original context, further mitigating the error propagation. To showcase it, we evaluate the accuracy of SteerPrompt to identify the gold sentences on the NQ dataset with Vicuna-7B and LLAMA3-8B-Instruct.

Table 4: Accuracy of gold sentence identification by SteerPrompt

| Vicuna-7B | LLAMA3-8B-Instruct |
|---|---|
| 64.02% | 67.07% |

We can tell that SteerPrompt achieves much higher accuracy of key sentence selection compared to the models' question answering accuracy (e.g., 64.02% of gold sentence identification v.s. 5.88% NQ EM by Vicuna-7B). As such, SteerPrompt converts the challenging bottleneck of contextual comprehension into an easier problem of key-sentence selection in a steerable way. The 64.02% accuracy of key sentence selection achieved by SteerPrompt is sufficient for it to significantly improve models' QA performance as shown in Table 3.

## 5.2 COMPARISON BETWEEN PROFILING STRATEGIES

To illustrate the effectiveness of the coarse-to-fine profiling strategy introduced in Section 3.2, we evaluate several different profiling approaches as follows:

• Greedy search proposed by Zhang et al. (2024): This strategy involves selecting the top-$k$ heads from all the attention heads in the models. The evaluation times for this strategy is $L \times H$.

• Group search inspired by Ainslie et al. (2023): Here, 8 adjacent heads from one layer form a group. Then, we evaluate them group-wise, and select the top-$k$ head groups. The evaluate times for this strategy is $LH/8$.

• Coarse-to-fine search: This strategy initially selects the top-$l$ layers and then chooses the head set only from the heads within these layers. The evaluation times for this strategy is $L + lH$.

where $L$ is the number of layers, and $H$ is the number of attention heads per layer. We compare them with a Vicuna-7B (Chiang et al., 2023) that has 32 layers, and 32 heads per layer. The results in Table 5 show that coarse-to-fine profiling significantly outperforms all the other strategies while reducing the total evaluation times by $4.5\times$ compared to the original greedy search in Zhang et al. (2024).

Table 5: Performance of SteerPrompt on NQ with Vicuna-7B when searching effective attention heads with different strategies. "# Eval" refers to the total evaluations with the profiling set.

| Method | # Eval | EM | Token F1 |
|---|---|---|---|
| Baseline | N.A. | 8.13 | 33.79 |
| Greedy search all heads | 1,024 | 14.81 | 35.63 |
| Group search (size of 8) | 128 | 12.12 | 36.13 |
| Coarse-to-fine search | 224 | 15.41 | 38.59 |

## 5.3 PERFORMANCE OF STEERPROMPT WHEN RETRIEVING SEVERAL PASSAGES

In this work, our primary objective is to enhance model reading comprehension to the provided evidence, and the gold evidence is always provided. Nonetheless, in practical applications, a retriever may simultaneously supply multiple similar and relevant passages. To demonstrate the effectiveness of SteerPrompt in such scenarios, we utilize DRP (Karpukhin et al., 2020) to retrieve an additional four passages, each ranked within the top four in relevance scores to the question. Along with the gold evidence, these five passages are then presented to the model. SteerPrompt is tasked with automatically extracting and highlighting the key sentence from this set. Table 6 displays the results using Vicuna-7B on NQ. Although the inclusion of more noisy passages generally leads to a performance decline, we still observe consistent improvements with SteerPrompt, underscoring the effectiveness of our approach.

Table 6: Performance of SteerPrompt on NQ with Vicuna-7B when additional passages are provided.

| Model | Method | NQ with Gold Evidence | | NQ with 5 Evidence | |
|-------|--------|------|----------|------|----------|
| | | EM | Token F1 | EM | Token F1 |
| **Vicuna-7B** | Direct Prompting | 5.88 | 32.21 | 2.50 | 22.18 |
| | Iterative Prompting | 4.36 | 31.48 | 1.74 | 21.99 |
| | SteerPrompt_{in-domain profiling} | 15.41 | 38.59 | 6.82 | 25.60 |

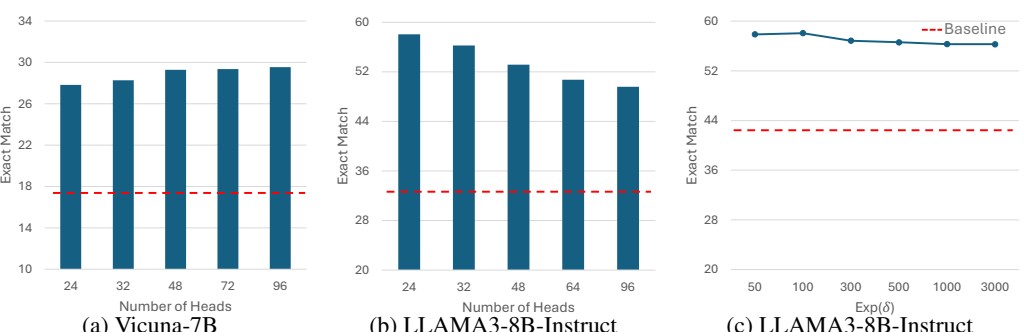

(a) Vicuna-7B      (b) LLAMA3-8B-Instruct      (c) LLAMA3-8B-Instruct

Figure 2: Ablation study of SteerPrompt performance on HotpotQA when steering different numbers of heads (2a and 2b) and setting different $\delta$ (2c). Dashed line in red refers to the baseline performance of direct prompting.

## 5.4 ABLATION STUDY

We conduct ablation study to discuss the performance of SteerPrompt given different number of attention heads for steering and different $\delta$.

**Varying the number of steered heads.** Figure 2a presents the performance variation of SteerPrompt with Vicuna-7B on HotpotQA dataset when steering different number of attention heads. Figure 2b illustrates the EM results for LLAMA3-8B-Instruct on the HotpotQA dataset under similar conditions. We see that steering more heads for SteerPrompt may result in slight performance degeneration, for example, the performance of LLAMA3-8B-Instruct on HotpotQA. This observation is similar to findings in previous work (see Figure 3 in Zhang et al. (2024)), where overemphasizing too many heads can lead models to focus on solely on highlighted information while ignoring other parts, potentially degenerating performance. In practice, we recommend applying SteerPrompt to steer a moderate number of heads. The optimal number of steered heads in our study is determined based on the performance metrics on the profiling data.

**Analyzing the sensitivity of $\delta$.** Figure 2c presents the sensitivity analysis for varying $\delta$ in (1) using LLAMA3-8B-Instruct on HotpotQA. We can see that the performance of SteerPrompt is not sensitive to the attention bias constant $\delta$. Changing its logarithm values (i.e., the scaling-down coefficient for non-highlighted tokens as elaborated in Appendix A) from 50 to 3000 does not induce dramatic performance variation. Therefore, we set $\delta$ as its default value $\log(100)$, which is the same as Zhang et al. (2024).

## 6 RELATED WORK

Large language models exhibit remarkable performance on (context-free) knowledge-intensive tasks, such as open-domain question answering (QA) (Kwiatkowski et al., 2019) and commonsense reasoning (Mihaylov et al., 2018; Clark et al., 2018), indicating that they encode substantial knowledge about open-world facts (Zhou et al., 2023) in their parameters. Despite their proficiency in memorization, different kinds of hallucinations in the output are observed, including factual knowledge hallucination (Huang et al., 2023; Yu et al., 2024), hallucination in summarization (Maynez et al., 2020; Pagnoni et al., 2021), hallucination in logical operations (Lyu et al., 2023; Huang et al., 2023). In this work, we focus on the factual knowledge hallucination due to models' unawareness of relevant knowledge or overlooking contextual information.

**Retrieval-augmented LLMs.** To address the problem of missing relevant knowledge, one popular method is to use retrieval-augmented LMs that supplement missing knowledge from external sources (Shi et al., 2023b; Peng et al., 2023). Retrieval augmentation requires that LLMs are sensitive to the input context and reliably understand the contexts. However, recent work shows that even if the relevant knowledge is presented, the model may still not fully comprehend the given evidence (Zhou et al., 2023; Yu et al., 2024; Wan et al., 2024).

**Prompt-based strategies.** To improve the reading comprehension of the models, various prompting strategies are designed to guide the model to detect the key information (Wei et al., 2022; Radhakr-ishnan et al., 2023), or focus on the given evidence (Zhou et al., 2023), while these extracted key information is only added as additional tokens in the input, and models still cannot grasp these new tokens.

**Model-based strategies.** Besides using prompting to improve the reading comprehension, some works augment an LLM's training data (Köksal et al., 2023; Hu et al., 2024; Chen et al., 2024). Alternatively, Shi et al. (2023a) proposes a context-aware decoding method to downweight the output distribution associated with the model's prior knowledge.

To the best of our knowledge, we are the first work to integrate key information prompting and explicit token highlighting during inference without any additional training.

## 7 CONCLUSION

In this paper, we address the challenge of contextual reading comprehension in open-book QA tasks and introduce SteerPrompt, an inference-only method that automatically identifies crucial information pieces within contexts and explicitly highlights them through steering a model's attention scores. SteerPrompt guides the model to focus on the essential information within contexts, leading to substantially improved model reading comprehension and performance. Remarkably, by integrate iterative prompting and attention steering techniques, SteerPrompt synergistically combines their advantages while mitigating their respective limitations.

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

# A DERIVATION FOR EQUATION 1

In this section, we present the derivation to show why (1) is equivalent to equation (2) in Zhang et al. (2024).

For the token that are not highlighted $j \notin \mathcal{G}$, Zhang et al. (2024) downweight their attention scores by scaling down their scores post-softmax by a coefficient $\alpha$ ($0 \le \alpha \le 1$): $\alpha \cdot \text{Softmax}(\boldsymbol{A}_{i\cdot})_j / C_i$ where $C_i = \sum_{j \in \mathcal{G}} \text{Softmax}(\boldsymbol{A}_{i\cdot})_j + \sum_{j \notin \mathcal{G}} \alpha \cdot \text{Softmax}(\boldsymbol{A}_{i\cdot})_j$. Now we show that:

$$\alpha \cdot \text{Softmax}(\boldsymbol{A}_{i\cdot})_j / C_i = \frac{\alpha}{C_i} \frac{\exp(\boldsymbol{A}_{ij})}{\sum_{j'} \exp(\boldsymbol{A}_{ij'})} \tag{5}$$

$$= \frac{\exp(\boldsymbol{A}_{ij} + \log(\alpha))}{C_i \sum_{j'} \exp(\boldsymbol{A}_{ij'})} \tag{6}$$

For the tokens in $\mathcal{G}$:

$$\text{Softmax}(\boldsymbol{A}_{i\cdot})_j / C_i = \frac{\exp(\boldsymbol{A}_{ij})}{C_i \sum_{j'} \exp(\boldsymbol{A}_{ij'})} \tag{7}$$

Therefore, after the renormalization, it is equivalent to condut the $\text{softmax}$ among $\boldsymbol{A}_{ij} + \log(\alpha)$ for $j \notin \mathcal{G}$ and $\boldsymbol{A}_{ij}$ for $j \in \mathcal{G}$, which is our simplified equation in (1).

# B EVALUATION DETAILS

## B.1 DATASET STATISTICS

|  | Profiling | Test |
|---|---|---|
| Natural Questions | 1,000 | 6,189 |
| HotpotQA | 1,000 | 4,190 |

Table 7: Natural Questions and HotpotQA data statistics after the preprocessing.

## B.2 THE DETAILED NUMBER OF ATTENTION HEADS FOR STEERING

| Model | NQ | HotpotQA |
|---|---|---|
| **Vicuna-7B** | top 64 heads from top 4 layers | top 96 heads from top 6 layers |
| **LLAMA3-8B** | top24 heads, 4 from each of top 6 layers | top24 heads, 4 from each of top 6 layers |
| **LLAMA3-70B** | top20 heads, 4 from each of top 5 layers | top 64 heads from top 5 layers |

Table 8: The detailed number of attention heads for steering

## C   PROMPT TEMPLATE DETAILS

**Prompt Templates of Two-Round Iterative Prompting**

[First Round]: A question, and a passage are shown below. Please select the key sentence in the passage that supports to answer the question correctly. Only output the exactly same sentence from the passage without other additional words.

Question: {Question}

Passage: {Evidence}

Sentence: _______________________________________________

[Second Round]: Answer the question below, paired with a context that provides background knowledge, and a key sentence. Only output the answer without other context words.

Context: {Evidence}

Key Sentence: {Predicted key sentence}

Question: {Question}

Answer:

