# OpenReview forum: "Model Tells Itself Where to Attend: Steerable Prompting for Reliable Reading Comprehension of LLM"
_ICLR.cc/2025/Conference — ICLR 2025 Conference Withdrawn Submission_

### Official Review · Reviewer_Z18A · 2024-10-31

**Soundness:** 3
**Presentation:** 3
**Contribution:** 2
**Rating:** 5
**Confidence:** 3

**Summary:**

This paper proposes SteerPrompt, which explicitly masks the attention weights of irrelevant tokens in reading comprehension tasks. Given a context and a question, the authors argue that only a small portion of the context is necessary to answer the question, whereas language models typically do not fully ignore the influence of irrelevant sentences through naive prompting. To address this, they add a negative value to the attention weights of irrelevant tokens before applying softmax, thereby forcibly reducing attention probabilities. This masking is applied on specific layers and attention heads, with the optimal layers or heads located via a greedy search.

**Strengths:**

1.	SteerPrompt appears well-suited for contextual searches, effectively reducing the influence of irrelevant sentences.

2.	Extensive experiments on all components of SteerPrompt are conducted, with results showing that even out-of-domain, SteerPrompt demonstrates its advantages.

**Weaknesses:**

1.	Innovation is slightly limited. The overall mechanism of SteerPrompt can be succinctly described as iterative prompting + PASTA.

2.	See major and minor questions for additional comments.

**Questions:**

Major Questions:

1.	In Table 3, a more direct comparison is missing, namely, a prompt that replaces the context with only the selected sentence. This method would allow a comparison of the effects between necessary sentences and the whole context, highlighting the impact of necessary sentences.

2.	Previous work on reading comprehension has mentioned cases where the golden answer is not provided in the context. Does SteerPrompt generalize to such cases? Specifically, will the model respond with “Not given” if the attention weights of all tokens are masked?

3.	It seems that only one key sentence is selected during the key sentence identification stage, and you mention that the model might fail to locate the golden key sentence. Did you attempt to recall multiple key sentences during this stage? This approach could improve the recall rate of the golden key sentence, although it would also introduce irrelevant information, creating a trade-off. I am curious about the influence of the number of selected sentences.

Minor Questions:

Does the set of selected attention heads remain consistent across different benchmarks? It would be interesting to know if there is an attention head that is universal across most reading comprehension datasets.

---

### Official Review · Reviewer_WgcX · 2024-10-31

**Soundness:** 2
**Presentation:** 2
**Contribution:** 2
**Rating:** 3
**Confidence:** 4

**Summary:**

Due to the lengthy prompt causing the model's attention to diverge, this study builds on Zhang et al. (2024) by selecting attention heads for re-weighting, guiding the model's attention to relevant context in prompts.
In this paper, authors proposed a method for automatically extracting key points from the context in open-book QA tasks, and  introduced Coarse-to-fine Model Profiling to improve the efficiency of retrieval attention heads. Experimental validation is conducted on three open-source LLMs.

**Strengths:**

The paper builds on PASTA and I think there are two main innovations:

1. In response to the nature of open-book QA tasks, the method is designed to automatically extract key sentences from the contexts by LLMs in openbook QA.

2. The profiling has been improved to enhance the efficiency of retrieval attention heads.

**Weaknesses:**

**In the  method construction:**
1. There is no special improvement compared with PASTA (Zhang et al. (2024)) in this paper. I think this paper introduces PASTA into QA tasks, which is rough in improvement. Meanwhile, the need for human annotation for PASTA is emphasized several times in this paper. However, PASTA does not have special human annotation, which is more in line with the actual situation. For example, if a user asks a question with a strong emphasis (output in json format), it only emphasizes the part of the prompt that needs to be emphasized. For QA tasks, context becomes a part of prompts, which is a different task. No need for human annotation cannot constitute the innovation point of the paper.

2. In this paper, the LLM directly extracts the key sentence in the contexts is relatively simple, only requires a prompt, and the accuracy is not high - (64.02%/67.07%). The effect of this error propagation is reflected in the fact that Iterative prompting performs worse than Direct prompting, can this approach be promoted? Besides, is there only one key sentence per context as shown in the prompt setting?

3. The prompt setup adopted by the paper is more like a two-step prompt setup than an iterative prompt.

**In the experiments:**

4. Since there are golden sentences in NQ, there is a lack of PASTA in the baseline selection of the paper for comparison.
In addition, iterative prompting is less effective than direct on most settings. It seems that the extracted sentence of the first step affects the performance of the model and thus propagates the error.

5. How about the performance of steerprompt+gold sentences in Table 3, and is it better than identified key sentences?

6. Lacking some experimental analysis, the performance of SteerPrompt in llama3-70B is worse than that in llama3-8B. Does it mean that steerprompt is only applicable to LLMs with small parameters (7B/8B)?

**Questions:**

1. There are some errors in the writing of the paper, such as NA on line 244, the title of Table 4 without a full stop, and the wrong font in Figure 2.

**Details Of Ethics Concerns:**

There are no Ethics Concerns in the paper.

---

### Official Review · Reviewer_ckB5 · 2024-11-01

**Soundness:** 3
**Presentation:** 4
**Contribution:** 3
**Rating:** 6
**Confidence:** 4

**Summary:**

its reading comprehension abilities. This method is the combination of two prior methods: iterative prompting to identify key sentences and post-hoc attention steering. By combining them, the authors combine the best of both worlds and avoid their limitations (increasing the prompt length and manually annotating key sentences respectively). They conduct experiments on 2 question answering datasets (NQ and HotpotQA) and show consistent gains by their method. Interestingly, even with out-of-domain tuning of the hyper parameters of their method, it remains effective and outperforms the baselines. The questions they tackle are:
1. Does SteerPrompt improve reading comprehension in in-domain Open-Book QA?
2. Does SteerPrompt improve reading comprehension in out-of-domain Open-Book QA?
3. What is the importance of each component of SteerPrompt?
4. How effective is each profiling strategy?
5. Does SteerPrompt also work for RAG (retrieving passages)?

**Strengths:**

1. Simple and effective method.
2. It works as an inference-only method (no need for retraining, etc)
3. The experiments clearly answers most of their questions successfully.
4. They propose new strategies to select the attention heads to steer and make it more efficient.

**Weaknesses:**

1. The number of datasets is bit small (2), specially considering that both of them uses Wikipedia pages. This would also limit their claims on the out-of-domain profiling tuning (question #2 from summary)
2. Related to the above point, this work has a lot of potential for RAG methods. However, there is only 1 experiment on NQ for this setting, which imho would limit their answer to question #5.

**Questions:**

1. Analyzing the RAG setup more could further provide more evidence for their claims in questions #2 and#4. I believe the BEIR benchmark could provide the authors ideas of more datasets to experiment. For example, SciFact uses PubMed Articles, which could further prove the effectiveness of out-of-domain profiling.
2. Another dataset authors could use without RAG is ConditionalQA, where the documents are not Wikipedia pages, and it also contains annotated evidence. This could  be used to clearly answer question #2.


BEIR: A Heterogeneous Benchmark for Zero-shot Evaluation of Information Retrieval Models (Thakur et al., in Neurips 2021 Datasets and Benchmark Track)
ConditionalQA: A Complex Reading Comprehension Dataset with Conditional Answers (Sun et al., ACL 2022)

---

### Official Review · Reviewer_ZtXs · 2024-11-02

**Soundness:** 3
**Presentation:** 3
**Contribution:** 3
**Rating:** 6
**Confidence:** 4

**Summary:**

This paper introduces SteerPrompt, an inference-only method which can automatically identify key contextual sentences and highlight them through attention score manipulation to improve models’ reading comprehension and performance on open-book QA tasks. Based on the previous work PASTA, this paper integrates iterative prompting and attention steering, and design an efficient coarse-to-fine search scheme to reduces the searching overhead.

**Strengths:**

1.	Combining iterative prompting and attention steering can mitigate the individual shortcomings of the two methods, and prevent multi-step errors caused by tokens. It also does not require changing any model parameters.
2.	Coarse-to-fine search scheme reduces the searching overhead effectively and achieves a better performance.
3.	Detailed analysis and ablation experiments are conduced to confirm the effectiveness of each component, and additionally the parameter settings are also discussed.

**Weaknesses:**

1.	For baseline, the method is just compared with direct prompting, iterative prompting in experiment part and attention steering in analysis part. As shown in Table 2 and Table3, Iterative prompting shows minor improvements and sometimes performs even worse than direct prompting, while in Table 2 just highlighting the entire context via attention steering can improve upon direct prompting a lot, is it better to add an evaluation of highlighting the entire context in the main experiments?
2.	It seems a bit odd to calculate the average of the EM and F1 together in Table 2. Is it better to calculate their averages separately?

**Questions:**

refer to the comments

---

### Official Review · Reviewer_QYyJ · 2024-11-05

**Soundness:** 3
**Presentation:** 2
**Contribution:** 2
**Rating:** 3
**Confidence:** 3

**Summary:**

The paper introduces SteerPrompt, a method aimed at enhancing the reading comprehension capabilities of large language models in open-book question answering tasks. SteerPrompt identifies key contextual information and explicitly highlights it by manipulating attention scores, without altering the model's parameters or requiring additional training data. The method integrates iterative prompting and attention steering, automating the identification of key sentences and steering the model's focus towards them. Experiments conducted on the Natural Questions  and HotpotQA datasets demonstrate significant performance improvements over baseline prompting strategies.

**Strengths:**

- The integration of iterative prompting and attention steering is a creative solution to improve LLMs' comprehension without additional training, which is both resourceful and practical.
- The paper provides a thorough analysis of the effects of SteerPrompt's components and its sensitivity to different parameters, which adds depth to the understanding of the method.

**Weaknesses:**

- The evaluation is focused on open-book question answering tasks, and it's unclear how well SteerPrompt would perform in other NLP tasks or scenarios.
- The performance of SteerPrompt is upper-bounded by the accuracy of key sentence identification, which may not always be perfect, especially with more complex or ambiguous contexts.
- The paper does not extensively discuss potential biases introduced by the model, especially since the method relies on the model's own predictions to identify key sentences.

**Questions:**

- Could the authors provide differences in the performance of SteerPrompt on different types of questions, such as single-step vs. multi-step QA, and factual vs. inferential questions?
- Have the authors considered visualizing the decision-making process of SteerPrompt to enhance the model's interpretability?
- How does SteerPrompt perform with adversarial examples that are deliberately constructed to confuse the model?
- Do different LLMs require adjustments to SteerPrompt's strategies or parameters to achieve the best performance?
- Is the performance of SteerPrompt stable over long periods of inference? Are there any issues with performance degradation?
- How does the model ensure privacy and data security when dealing with sensitive data?

---

### Note · Authors · 2024-11-25

I have read and agree with the venue's withdrawal policy on behalf of myself and my co-authors.